# OpenReview forum: "Jailbreak Transferability Emerges from Shared Representations"
_ICLR.cc/2026/Conference — ICLR 2026 Poster_

### Official Review · Reviewer_aZNi · 2025-10-22

**Soundness:** 3
**Presentation:** 3
**Contribution:** 3
**Rating:** 8
**Confidence:** 4

**Summary:**

This work shows that jailbreak transferability is strongly tied to representation similarity. The author support this claim by showing a clear correlation between representation similarity and attack transferability, as well as performing an intervention and finetuning a student model through knowledge distillation and showing it becomes vulnerable to the same attacks as the teacher model.

**Strengths:**

- The paper advances our understanding of LLM jailbreaks and the cases where they transfer.
- The methodology is sound and clearly explained and the results support the authors' claims.
- The paper is well written, with motivation and contributions clearly communicated.

**Weaknesses:**

I did not find any major weaknesses in this paper, however while the claims of the paper are very well supported, they are also a bit narrow in that they connect jailbreak attacks to transferability but don't provide further insight on the precise mechanism or the implications for preventing jailbreaks. This keeps me from recommending spotlight/oral acceptance but I still think the paper is good contribution and can be accepted in its current form.

Overall, a more detailed discussion of the implications of this work and connections to previous work would improve the paper. For example fine tuning on safe data leading to vulnerability to the same jailbreaks could be seen as an example of subliminal learning (https://arxiv.org/abs/2507.14805). Making these kinds of connections and discussing them at least in the Appendix could be of interest to the readers.

**Questions:**

The difference between the transferability for persona-based and cypher-style jailbreaks is mentioned in the abstract and the introduction but then barely discussed outside of Appendix E. I would be interested in getting more of your thoughts on why we see this difference. Would it make sense to look at the representation similarity of different groups of jailbreaks? Are the representations of persona-based jailbreaks more similar across models than the representations of cypher-based jailbreaks?

---

> ### Author Response · Authors · 2025-11-19
>
> Thank you for the supportive review and your recognition of both the clarity and the methodological rigor.
>
> > The paper does not provide implications for preventing jailbreaks
>
> We provide a new section to the paper discussing what approaches we think are most promising based on our results.
>
> > Deeper discussion of mechanisms and implications.
>
> We agree that deeper discussion strengthens the paper and have added an extended discussion. Our primary contribution is isolating representational similarity as the central explanatory factor. Since we measure representational similarity over a large dataset of diverse prompts, we expect the mechanisms at play to vary widely. This makes it hard to attribute the alignment to specific mechanisms. That being said, we also believe that representational similarity necessarily requires similarity of the underlying mechanisms. We hope that future work studies this by, e.g., decomposing neural networks into circuits and studying their similarity.
> We also agree that there are connections to concurrent work which we highlight in the updated version. We believe subliminal learning is a particularly interesting example of this, since it has already been connected to emergent misalignment in prior work. We argue that our observations are similar in spirit and points to a broader safety concern: Fine-tuning language models on benign data can lead to side effects that undermine safety, even when trying to control for that (in our case, by including refusal examples from the student model into the fine-tuning dataset).
>
> > Differences between cipher-based and persona-based jailbreaks:
>
> We agree that the difference between persona-based and cipher-based jailbreaks deserves more explicit discussion in the main text.
> Persona-based jailbreaks use natural-language roles, conversational framing, and social cues. These features are learned robustly during pretraining and are represented similarly across model families. As a result, persona jailbreaks interact with parts of the latent space that are already highly aligned, which makes their success more likely to transfer.
> Cipher-based jailbreaks, in contrast, depend on tokenization artifacts, rare transformations, and brittle handling of unusual input strings. These behaviors vary widely across architectures and vocabularies, so cipher prompts tend to fall into representation regions that are not shared across models. This explains why their effectiveness drops sharply with scale and why they show little transfer, even when models are otherwise similar.
> While our representational similarity metrics are computed only on benign data, the observed asymmetry is consistent with our mechanism: the jailbreaks that transfer most reliably are precisely those that operate in the regions of representation space that models naturally share. Measuring jailbreak-specific embeddings could certainly be interesting, but the fact that persona attacks align with benign representational similarity strengthens the causal interpretation rather than requiring additional measurements.
> We clarify this reasoning and move more of the Appendix E discussion into the main text.

---

> > ### Comment · Reviewer_aZNi · 2025-11-27
> > **Response**
> >
> > Thank you for your comments and clarification, I will maintain my score and recommend acceptance of this paper.

---

### Official Review · Reviewer_byug · 2025-10-30

**Soundness:** 3
**Presentation:** 3
**Contribution:** 2
**Rating:** 4
**Confidence:** 3

**Summary:**

This paper discusses the reason behind jailbreak transferability in LLMs from the perspective of shared representations. The study employs benign data distillation techniques from a large model to a small model, and identifies two key factors: representational similarity and jailbreak strength.

**Strengths:**

1. Clear expression and fluent writing.
2. Rigorous evaluation using the StrongREJECT judge.
3. Aboundant experiments on open-weight models and different jailbreak attacks.
4. Validated the effectiveness of distillation techniques in enhancing the transferability of jailbreak prompts.

**Weaknesses:**

1. Many of the findings in this paper are quite basic and not new to me, as similar conclusions have appeared in other fields. For instance, distillation has been shown to improve model similarity [1], enhance black-box transferability [2-3], and boost robustness [4-5]. Additionally, a recent paper on LLM attacks found that distillation can improve the effectiveness of prompt attacks on smaller models [6], which overlaps significantly with the core conclusion of this paper. Personally, this paper feels more like an exploratory discussion of a phenomenon rather than a contribution to advancing the SOTA baselines, and the repetition of these conclusions is a significant flaw.
2. Lack of evaluation about transferability on commercial LLM APIs, such as GPT, deepseek, grok.
3. If the findings of this paper could form a specific attack methodology and presented as a dedicated section, followed by a direct comparison with more recent baselines, it would enhance the completeness of the paper.

[1] Similarity-Preserving Knowledge Distillation
[2] Teach Me to Trick: Exploring Adversarial Transferability via Knowledge Distillation
[3] Improving the Transferability of Adversarial Examples by Inverse Knowledge Distillation
[4] Improving Adversarial Robustness Through Adaptive Learning-Driven Multi-Teacher Knowledge Distillation
[5] ProARD: progressive adversarial robustness distillation: provide wide range of robust students
[6] Efficient and Stealthy Jailbreak Attacks via Adversarial Prompt Distillation from LLMs to SLMs

Above all, my initial score is 4, and it may change depending on the rebuttal.

**Questions:**

Perhaps it could be worth trying to distill a single model from the ensemble of multiple models?

---

> ### Author Response · Authors · 2025-11-19
>
> Thank you for your review and your recognition of both the clarity and the methodological rigor.
>
> > Findings are basic and not new.
>
> We appreciate the concern about novelty, and we agree that distillation and transfer phenomena have been explored in a variety of settings. However, we believe our work addresses a substantially different problem domain, and we would like to clarify the distinction.
> The works cited in [1]-[5] focus on classical adversarial examples in computer vision (primarily image classification) and study how distillation affects robustness and transferability in that setting. As shown in recent work (Gupta et al., 2025), several intuitions from vision-based adversarial examples do not carry over to jailbreak attacks on large language models. Because the threat models, architectures, and failure modes differ significantly, we see our results as nontrivial and not directly implied by those findings.
> Regarding [6], the goals and conclusions of that work differ from ours in important ways. Their approach distills Llama3.2-1B into BERT to create a stronger attack generator. In contrast, our paper investigates how distillation among victim models affects inter-model similarity and the resulting transferability of jailbreak prompts. This causal link (showing that distillation between victims increases jailbreak transferability) does not appear to be established in prior literature. While [6] is an interesting result, we see it as complementary rather than overlapping.
> To the best of our knowledge, our work provides the first systematic evidence of how distillation reshapes jailbreak vulnerability and transferability specifically in LLMs. We believe these findings fill an important gap in understanding AI safety and security.
> > This paper feels more like an exploratory discussion of a phenomenon rather than a contribution to advancing the SOTA baselines.
>
> Our work provides a complementary contribution to work that advances SOTA attack baselines. We provide fundamental insight into why jailbreaks transfer, and in turn, how jailbreaks work. We believe that our contributions provide intuition and evidence about the dynamics of jailbreaks and are very valuable to the AI safety and security communities in their own right. We provide an additional discussion of the implications for both jailbreak attacks and defenses in the paper.
>
> > Lack of evaluation about transferability on commercial LLM APIs, such as GPT, deepseek, grok.
>
> We intentionally focused on open-weight models for methodological rigor and reproducibility. Black-box models accessed via APIs are regularly updated and often are not even a single model (e.g. due to model routing), making exact reproduction difficult, and they lack the white-box access necessary to compute our core model representational similarity. While one could adapt our approach into an attack on proprietary closed-weight models, the lack of transparency would make it difficult to attribute causes of success or failure to such an attack.
>
> > If the findings of this paper could form a specific attack methodology and presented as a dedicated section, followed by a direct comparison with more recent baselines, it would enhance the completeness of the paper.
>
> We believe this work informs defense methodology. We provide a new section to the paper discussing what approaches we think are most promising based on our results.
>
>
>
> *References:*
>
> Gupta, I., Schaeffer, R., Kazdan, J., Liu, K. and Koyejo, S., 2025. Understanding Adversarial Transfer: Why Representation-Space Attacks Fail Where Data-Space Attacks Succeed.
>
> Zou, A., Wang, Z., Carlini, N., Nasr, M., Kolter, J.Z. and Fredrikson, M., 2023. Universal and transferable adversarial attacks on aligned language models.

---

> > ### Comment · Reviewer_byug · 2025-11-22
> >
> > My main issue has been resolved: the difference between this method and [6], and why it doesn't attack the API. Overall, the author has made significant experimental efforts and analyses. However, as an analytical article, the core findings are still not new enough for me. So, I decide to raise my rate.

---

> > > ### Author Response · Authors · 2025-11-22
> > >
> > > Are there any other questions you have? Did you want to raise your score to a 6?

---

> > > > ### Author Response · Authors · 2025-11-28
> > > >
> > > > Thank you for raising your score to a 6. We appreciate the feedback.

---

### Official Review · Reviewer_pGjz · 2025-10-31

**Soundness:** 3
**Presentation:** 3
**Contribution:** 3
**Rating:** 6
**Confidence:** 4

**Summary:**

This paper investigates the underlying mechanisms driving the *transferability of adversarial jailbreak prompts* across large language models (LLMs). By systematically evaluating 20 open-weight models and 33 attack templates, the authors identify two principal factors predicting transfer: (1) representational similarity under benign prompts and (2) jailbreak strength on the source model. Through a causal intervention—benign-only distillation—the authors demonstrate that increasing representational similarity causally enhances transferability, supporting the hypothesis that shared internal representations, rather than safety-tuning artifacts, underlie this phenomenon. The study represents the most comprehensive empirical exploration of jailbreak transfer to date and proposes a clean, reproducible experimental paradigm for future research.

**Strengths:**

1. The study compiles the largest known dataset of jailbreak transfer evaluations (20 models × 33 attacks × 313 prompts).
2. The authors carefully control for attack strength, preventing the common confounding that “stronger attacks transfer more.”
3. The benign-only distillation protocol provides a safe, controlled method to causally test representational effects without ethical concerns.
4. The results offer a clear practical implication for defense research: surface-level alignment is insufficient; external classifier-based safety layers may remain necessary.

**Weaknesses:**

1. Limited scope of distillation: Only three teacher–student pairs, trained for one epoch under a single hyperparameter configuration.
2. Limited practical impact: Even after distillation, transfer rates in large models remain low (≤10% in Fig. 6), somewhat constraining real-world significance.

**Questions:**

1.  In Figure 5, after epoch ≈ 0.3 the similarity curve in Figure 4 has almost saturated, yet the mean transfer score shows a mild downward trend toward epoch 1. What causes this decoupling? Could you provide results for two or three additional epochs to clarify whether transferability plateaus or continues to drop as the model further overfits to benign responses?
2. Beyond empirical correlation, can your representational-alignment hypothesis inspire new attack or defense mechanisms? For instance, if jailbreak transfer arises from shared latent geometry, could one design targeted perturbations or representation-level regularizers to either exploit or mitigate this alignment? As it stands, the theoretical framing feels somewhat descriptive rather than generative—clarifying its actionable implications would strengthen the paper’s overall significance. This is particularly important given that, as shown in the figures, the absolute transferability remains quite low even after distillation, which raises questions about the practical impact of the proposed explanation.

---

> ### Author Response · Authors · 2025-11-19
>
> We appreciate your careful review and your recognition of the scale of the analysis and the importance of controlling for attack strength.
> > Limited scope of distillation
>
>  We agree that, compared to the large-scale correlational study, the intervention analysis necessarily uses a smaller set of teacher-student configurations. This choice reflects a deliberate trade-off: rather than running many closely related experiments, we selected three teacher-student pairs spanning different model families (Llama, Gemma, Qwen) and different scales (3B-14B). This ensures that the intervention is tested across different architectures, training data, and objectives. This is important since we expect those factors to most strongly affect global representation geometry. Additionally, recall that we are doing full fine-tuning, so running distillation on models up to 14B parameters is also quite computationally expensive (the 14B parameter student model required 8xH200 GPUs, resulting in >1TB of VRAM). We prioritized depth (i.e. full fine-tuning) and cross-family coverage over breadth (i.e. dozens of similar pairs).
> Finally, the distillation experiment is not meant to stand alone. It serves as a causal test that complements the large-scale correlational evidence across 380 model pairs. The fact that both analyses point to the same mechanism (representation similarity as the driver of transfer) provides strong evidence despite the necessarily smaller intervention set.
>
> > Limited practical impact
>
> We appreciate the reviewer’s concern that the absolute transferability values remain modest (Figure 6(c) shows ≤10% ASR). Our goal in this work is not to demonstrate a high-impact jailbreak attack, but to explain the mechanism by which transferability emerges. Our results show that representational similarity is a strong predictor of transferability. When combined with recent work proposing the Platonic Representation Hypothesis (that sufficiently scaled models converge toward shared latent structures) our findings imply that transferability may increase over time even as robustness increases, because shared latent representations become more similar as models scale.
> The main reason absolute transferability is low is that cipher-based jailbreaks (ROT13, Base64, GCG prompts) fail almost entirely on larger aligned models (e.g. Gemma2-27B is not very susceptible to GCG attacks). In contrast, persona-based jailbreaks remain effective and, crucially, are the most transferable across models. Furthermore, Gemma2-27B and Qwen2.5-14B have very low model similarity (see Figure 4(c)) to begin with and we are able to increase similarity quite a bit, but not as much as the other pairs. Regardless, we still see an increase in model similarity which causal increases transferability of GCG style attacks.
>
> > Why does transferability slightly decline after ~0.3 epochs even though similarity has saturated?
>
> The decoupling is most noticeable in Figure 5(b). In this specific case, we observe that benign-only distillation strengthens refusal for some jailbreak types, namely AIM (Figures 7 & 8). This means that benign-only distillation can actually increase safety. Since the metric is averaged across jailbreak types, an increase in refusal to those jailbreak types could pull the overall curve down a bit. Overall, since the distillation objective does not explicitly optimize for jailbreak transferability, mild non-monotonicity is expected. We clarify this in the revised text.
>
> > Do our results inspire defense mechanisms?
>
> We provide a new section to the paper discussing what approaches we think are most promising based on our results.

---

### Official Review · Reviewer_VGTQ · 2025-11-06

**Soundness:** 3
**Presentation:** 4
**Contribution:** 3
**Rating:** 8
**Confidence:** 3

**Summary:**

This paper argues that jailbreak transferability arises from shared internal representations between language models. To provide causal evidence, they use benign-only distillation to increase a student model's representational similarity to a teacher, which in turn makes the student more vulnerable to the teacher's jailbreaks. Experimental results demonstrate that jailbreak transfer as a fundamental consequence of representational alignment, suggesting that models that think alike inherit each other's vulnerabilities.

**Strengths:**

1.This paper provides a new and important perspective on jailbreak transfer, framing it as a fundamental consequence of shared representations.

2.The authors' novel distillation experiment provides powerful causal evidence for their claims, which is a very convincing and methodologically sound approach.

3.The large-scale quantitative analysis across 20 models makes the observed correlation between similarity and transferability highly robust and credible.

**Weaknesses:**

1. The paper's causal claim relies on the assumption that benign-only distillation solely increases representational similarity. However, this fine-tuning process could also inadvertently degrade the student model's general capabilities and original safety training effectiveness, which could also explain its increased vulnerability.

2. The study's conclusions are dependent on the chosen mutual k-nearest neighbors metric for measuring similarity. Could the authors explain why they chose this metric or demonstrate whether this strong correlation would persist with other similarity metrics?

**Questions:**

see Weaknesses

---

> ### Author Response · Authors · 2025-11-19
>
> Thank you for your positive review and your recognition of the methodological rigor.
>
> >  Does benign-only distillation degrade the student model and inflate vulnerability?
>
> This is definitely a concern that we address at the beginning of Section 5. In the experimental setup paragraph early in Section 5, we reference previous work that makes this point (Wolf et al., 2024; Qi et al., 2023; He et al., 2024), and then, discuss how we mitigate this concern. Specifically, we sample refusal responses to prompts for AdvBench from the student model and mix these in with the benign prompt-response pairs from the teacher. We find that this maintains the models’ global safety in general. Evidence of this is shown in Figures 7 & 8, where the Qwen2.5-7B model (one of the most unsafe models) is distilled from Llama-3.1-8B (one of the safest models) and actually becomes much safer after distillation.
>
> > Why mutual k-NN? Does the correlation persist with other similarity metrics?
>
> We chose this metric because it directly reflects structural alignment of latent geometry, which is the central point of our hypothesis. Specifically, it has the following important properties: (1) it measures topological similarity of the representation space (not absolute distance), (2) it is invariant to the model’s hidden dimension, scaling, and rotation, which might differ widely across models, and (3) it focuses on the structural neighborhood relationships that determine how prompts cluster semantically.
> We agree that other metrics could also be informative. Our goal in this paper was not to exhaustively benchmark metrics, but to show that one robust, topologically-aware metric strongly predicts transferability across 380 model pairs.

---

> > ### Comment · Reviewer_VGTQ · 2025-11-28
> >
> > Thank you for your detailed response. All my questions have been resolved. I think this paper makes a valuable contribution to understanding jailbreak transferability, and I will maintain my score.

---

### Author Response · Authors · 2025-11-19

We thank all reviewers for their thoughtful and constructive feedback. We appreciate that the reviewers found the paper well-written, methodologically sound, and that several reviewers highlighted the novelty and importance of our central finding, supported by both large-scale correlational evidence and a causal distillation intervention.

---

### Meta-Review · Area_Chair_uys5 · 2026-01-06

**Summary:**

This paper investigates the question of jailbreak transferability. The authors show that distillation on benign data increases the transferability of jailbreaks as well as representation similarity.

Overall, the reviews have been very positive.

I recommend this paper for a poster since no reviewer recommended this paper for an oral (and there is no spotlight this year)

I would recommend that the author mitigate the claim made in the title and the abstract regarding the causal relationship between representation and jailbreak transferability "we show that deliberately increasing similarity through benign-only distillation causally increases transfer". To the extent of my understanding, your experiment shows that distillation increases transferability (your treatment is the distillation). Similarity could merely be correlated with transfer without being causal.

**Reviewer Concerns:**

The reviewers engaged in the discussion with the author. No concerns remained after the discussion.

**Reviewer Scores:**

They would have remained the same.

---

### Decision · Program_Chairs · 2026-01-26

Accept (Poster)